# The Future of Stress Management: Integration of Smartwatches and HRV Technology

**DOI:** 10.3390/s23177314

**Published:** 2023-08-22

**Authors:** Ravinder Jerath, Mohammad Syam, Shajia Ahmed

**Affiliations:** Mind Body Technology, Augusta, GA 30907, USA

**Keywords:** smartwatch, stress, wearable device, heart rate variability, comparative analysis

## Abstract

In the modern world, stress has become a pervasive concern that affects individuals’ physical and mental well-being. To address this issue, many wearable devices have emerged as potential tools for stress detection and management by measuring heart rate, heart rate variability (HRV), and various metrics related to it. This literature review aims to provide a comprehensive analysis of existing research on HRV tracking and biofeedback using smartwatches pairing with reliable 3rd party mobile apps like Elite HRV, Welltory, and HRV4Training specifically designed for stress detection and management. We apply various algorithms and methodologies employed for HRV analysis and stress detection including time-domain, frequency-domain, and non-linear analysis techniques. Prominent smartwatches, such as Apple Watch, Garmin, Fitbit, Polar, and Samsung Galaxy Watch, are evaluated based on their HRV measurement accuracy, data quality, sensor technology, and integration with stress management features. We describe the efficacy of smartwatches in providing real-time stress feedback, personalized stress management interventions, and promoting overall well-being. To assist researchers, doctors, and developers with using smartwatch technology to address stress and promote holistic well-being, we discuss the data’s advantages and limitations, future developments, and the significance of user-centered design and personalized interventions.

## 1. Introduction

Over the past decade, wearable technology has gained significant traction, with smartwatches emerging as one of the most popular and widely adopted devices in this category. A smartwatch is a wrist-worn device equipped with various sensors, connectivity features, and a display screen, offering functionalities beyond traditional timekeeping [1]. These devices have seen exponential growth in popularity due to their versatility, convenience, and ability to seamlessly integrate with smartphones and other smart devices. Smartwatches initially gained attention for their fitness and activity tracking capabilities. They provided users with the ability to monitor their physical activity, track steps, measure heart rate, and calculate calorie expenditure [2]. The incorporation of advanced sensors, such as optical heart rate sensors and accelerometers, allowed users to gain insights into their health and wellness in real time. Beyond fitness tracking, smartwatches quickly expanded their features and functionalities to include communication, notifications, mobile apps, music playback, and more [3].

As the capabilities of smartwatches advanced, manufacturers recognized the potential to address another pressing issue affecting individuals’ well-being: stress management. The detrimental effects of chronic stress on physical and mental health have led to a growing demand for tools and techniques that help individuals monitor and alleviate stress levels [4]. This is where the integration of heart rate variability (HRV) analysis into smartwatches has garnered attention. HRV, which measures the variation in time intervals between consecutive heartbeats, is an indicator of autonomic nervous system activity [5,6]. It has been widely studied and recognized as a valuable metric for assessing stress levels, emotional states, and overall well-being. By exploiting optical sensors and advanced algorithms, smartwatches can capture and analyze HRV data, providing users with insights into their stress levels and offering interventions to manage and reduce stress effectively [7,8]. The integration of stress management features based on HRV analysis has positioned smartwatches as holistic wellness devices [8]. By combining fitness tracking, communication, and stress management capabilities, these devices have the potential to empower individuals in maintaining a healthy lifestyle and have emerged as a promising solution at the intersection of technology and personal health.

Heart rate variability (HRV) refers to the fluctuation in the time intervals between consecutive heartbeats, also known as R-R intervals, as measured by electrocardiography (ECG) or optical sensors [5,6]. It reflects the dynamic balance between the sympathetic and parasympathetic branches of the autonomic nervous system (ANS), which regulates our body’s physiological responses to stressors [9]. The ANS plays a crucial role in modulating stress by regulating heart rate, blood pressure, respiration, and other vital functions. The sympathetic branch of the ANS is responsible for the “fight-or-flight” response, activating the body to cope with stress, while the parasympathetic branch promotes relaxation and restoration [3,10].

When individuals experience acute or chronic stress, the sympathetic branch of the ANS becomes dominant, leading to increased heart rate and decreased HRV. Conversely, during periods of relaxation and recovery, the parasympathetic branch prevails, resulting in decreased heart rate and increased HRV [9,11]. Therefore, a higher HRV is generally associated with a more adaptive stress response and better overall well-being [9,12].

HRV analysis provides valuable insights into an individual’s physiological state, including their stress levels, emotional states, and autonomic balance [11]. By continuously tracking HRV throughout the day, these devices can provide real-time feedback on stress levels and suggest personalized interventions to help users regulate their stress response [8]. These interventions may include breathing exercises, guided meditations, mindfulness prompts, or activity recommendations tailored to everyone’s needs. Furthermore, comparing HRV patterns before and after implementing stress reduction techniques, individuals can objectively assess the impact of different strategies and make informed decisions about which methods work best for them and help them to enhance their self-awareness, adopt healthier coping mechanisms, and ultimately lead a more balanced and stress-resilient life [13,14].

The purpose of conducting a literature review comparing smartwatches in managing stress using heart rate variability (HRV) is to systematically analyze and synthesize existing research and knowledge on the effectiveness of different smartwatches as stress management tools. The review aims to compare the capabilities, features, and performance of smartwatches in measuring HRV and providing stress reduction interventions. By examining the literature, the review intends to identify the strengths, limitations, and gaps in the current body of knowledge to offer insights for future research, development, and practical application of smartwatches in stress management.

## 2. HRV and Stress Management

Heart rate variability (HRV) relates to the variation in the time intervals between successive heartbeats. It is a physiological phenomenon caused by the balance of the sympathetic and parasympathetic branches of the autonomic nervous system. Heart rate variability (HRV) has gained a lot of interest as a stress and overall well-being metric. Here is a quick rundown of heart rate variability (HRV) as a stress indicator:

The autonomic nervous system (ANS) controls various involuntary bodily functions, including heart rate. The sympathetic nervous system (SNS) handles the “fight-or-flight” response, whereas the parasympathetic nervous system (PNS) controls the “rest-and-digest” response. HRV depicts the dynamic interaction between these two branches. High and low HRV indicate greater variability in heartbeat intervals, indicating a flexible and adaptable autonomic nervous system (ANS) [11]. It suggests a stronger parasympathetic response and increased stress resilience. Lower heart rate variability (HRV), on the other hand, indicates decreased variability and increased sympathetic dominance, which may be associated with chronic stress, fatigue, or health concerns.

Stress activates the sympathetic nervous system (SNS), resulting in an increase in heart rate and a decrease in heart rate variability (HRV). Chronic stress can disrupt the balance of the sympathetic nervous system (SNS) and parasympathetic nervous system (PNS), resulting in lower total heart rate variability (HRV). Prolonged sympathetic dominance may be associated with a number of health problems, including cardiovascular disease, anxiety, and depression [5,9]. Electrocardiography (ECG or EKG), which records the electrical activity of the heart, is often used to measure heart rate variability (HRV). The recorded data are then analyzed to produce HRV features such as time-domain and frequency-domain measures, which provide information regarding heart rate autonomic modulation.

HRV biofeedback is a technique for increasing HRV and managing stress. It is a much more active and controlled stimulation of the neurological system to create changes in baroreceptor reflex, respiratory sinus arrythmia, and other physiological systems that connect to the lungs, cardiovascular system, and brain system. Giving participants real-time feedback on their HRV patterns and supporting them in acquiring self-regulation strategies to enhance parasympathetic activity and decrease sympathetic arousal are part of it [15]. HRV biofeedback has been used to help with stress management, anxiety control, and performance enhancement.

Individuals’ HRV levels and stress reactivity differ. Age, fitness level, genetics, lifestyle, and underlying health conditions are just a few of the variables that can influence HRV [9,16]. As a result, determining an individual’s baseline HRV and taking these factors into consideration when interpreting HRV values is crucial.

Monitoring heart rate variability (HRV) can help with stress management for several reasons:

HRV provides information on the physiological response of the organs to stress. Individuals can detect early signs of stress and intervene before they worsen by regularly monitoring their HRV [9,17]. The early detection of stress allows for early intervention and prevention. In addition, HRV monitoring allows for a customized stress assessment. Everyone has a unique baseline HRV, and variations from this baseline can indicate changes in stress levels [9]. By researching their own patterns, people can gain a better knowledge of their stress reactions and develop individualized stress management strategies. HRV biofeedback training comprises learning self-regulation skills with the use of real-time HRV readings [18,19]. Individuals can improve their ability to self-regulate physiological responses by assessing the effects of stressors on HRV and using stress-reduction strategies. Biofeedback training allows people to actively manage their stress levels and achieve a condition of balance and relaxation.

Chronic stress can be harmful to both physical and mental health. Monitoring HRV can help people recognize stressors and make healthy lifestyle adjustments. Individuals can improve their stress management, resilience, and overall well-being by actively regulating their stress levels and boosting their HRV [9,19].

HRV monitoring can provide useful information on the efficacy of lifestyle adjustments. Individuals could examine how changes in sleep patterns, exercise routines, food, or relaxation techniques affect their HRV [20,21]. This feedback loop allows for the ongoing optimization of stress management strategies based on individual responses and preferences. As a result, HRV monitoring contributes to a holistic approach to health by taking the mind–body link into account. Stress influences both mental and physical health, and HRV serves as a bridge between the two. By monitoring HRV, people can actively manage stress from all angles and increase their overall health and fitness.

The concept of the autonomic nervous system (ANS) and its role in stress response and control can be used to comprehend a theoretical framework that links HRV and stress reduction. The theoretical framework described below describes the relationship between HRV and stress reduction:

The ANS controls the body’s reaction to stress. Its two branches are the sympathetic nervous system (SNS) and the parasympathetic nervous system (PNS). In stressful situations, the sympathetic nervous system (SNS) triggers the “fight-or-flight” response, resulting in increased heart rate, blood pressure, and decreased HRV. The parasympathetic nervous system (PNS), on the other hand, activates the “rest-and-digest” response, promoting relaxation, a lower heart rate, and an increase in HRV [3,6,9,10]. Again, the dynamic interaction of the sympathetic nervous system (SNS) and parasympathetic nervous system (PNS) is reflected in HRV. Higher HRV indicates a more balanced and adaptable autonomic nervous system (ANS), with the parasympathetic nervous system (PNS) playing a larger role. Lower HRV, on the other hand, implies sympathetic nervous system (SNS) dominance and decreased stress adaptation [6,9,10]. Various stress reduction techniques attempt to reduce autonomic nervous system (ANS) activity and increase parasympathetic nervous system (PNS) dominance. These treatments include deep breathing techniques, meditation, mindfulness, progressive muscle relaxation, and biofeedback training [22,23].

Stress reduction measures can improve HRV. Individuals who adopt these methods activate the parasympathetic nervous system (PNS), resulting in relaxation and improved HRV. Approaches to stress reduction that frequently activate the parasympathetic nervous system (PNS) can result in long-term changes in autonomic nervous system (ANS) balance and improved stress resilience [9]. The connection between HRV and stress reduction results in a positive feedback loop. Individuals who utilize stress-reduction techniques and perceive an increase in HRV receive positive feedback [13]. This feedback loop encourages continual stress-reduction practice, resulting in greater improvements in HRV and stress reduction. Consequently, stress resilience can be promoted by the regular practice of stress reduction techniques as well as subsequent increases in HRV. Individuals who have a healthy ANS response become better equipped to handle stress and preserve emotional well-being [24]. Increased HRV shows a flexible and adaptive ANS capable of responding to stimuli effectively without excessive physiological arousal [11].

Thus, this theoretical framework provides a conceptual understanding of the link between HRV and stress reduction. Because the actual efficacy and effects of stress reduction approaches on HRV differ from person to person, empirical research and therapeutic support should be used in tandem with this paradigm.

## 3. Smartwatches and Stress Management

Smartwatches are popular wearable devices to help reduce stress [25]. Smartwatches’ stress-reduction potential is discussed below.

Most smartwatches monitor the wearer’s heart rate throughout the day. Since stress and anxiety raise the heart rate, smartwatches can show stress levels. Real-time heart rate data may reveal stress causes [8]. Advanced smartwatches analyze HRV. HRV measures the fluctuation in heartbeat intervals to better assess stress and autonomic nervous system balance. HRV-analysis smartwatches may help customers manage stress [5,9].

Many smartwatches track stress using heart rate, HRV, and other metrics. These devices may detect high stress levels and provide real-time reminders to do breathing exercises or mindfulness [8]. Some smartwatches include relaxing or breathing techniques. These features guide users through deep breathing or mindfulness exercises, helping them relax in stressful times. The watches may give visual or tactile feedback to help users follow the workouts [26].

Sleeping sufficiently reduces stress. Smartwatches measure sleep duration, phases, and quality. Examining sleep patterns may disclose causes of stress and tiredness [27]. Exercise reduces stress. Smartwatches measure steps, distance, calories, and exercise. Smartwatches may reduce stress by encouraging regular activity and providing feedback [28,29].

Smartwatches can plan mindfulness or relaxation breaks. These reminders may encourage users to take short breaks, breathe deeply, or meditate for stress management and mental health [30]. Smartwatches also link to smartphone applications or online platforms for extensive data analysis. Users may analyze stress patterns, trends, and historical data to discover triggers and make educated lifestyle adjustments and stress reduction choices [31].

Smartwatches detect and analyze HRV using optical heart rate sensors and advanced algorithms. Smartwatches assess heart rate variability: Smartwatches’ undersides include optical heart rate sensors that touch the wearer’s skin. LEDs illuminate the skin, while photodiodes detect the reflected light. Photoplethysmography (PPG) is this technique. LEDs send light into the skin, which the blood vessels absorb, and the photodiodes reflect. Photodiodes detect blood-induced light intensity changes. The wearer’s pulse is shown as a PPG signal [32]. Smartwatches use the PPG signal to calculate heart rate by analyzing the period between heartbeats. Smartwatches display real-time heart rate data [33]. Algorithms analyze the raw PPG signal and derive HRV data in smartwatches. HRV monitoring relies on variations in pulse intervals [34].

Frequency-domain HRV analysis is commonplace. The FFT converts the raw PPG signal into the frequency domain. This transformation isolates signal frequency components [5,35]. Frequency-domain analysis determines HRV parameters. High frequency (HF), low frequency (LF), and the LF/HF ratio are factors. High-frequency (HF) power represents the parasympathetic nervous system, LF power represents sympathetic and parasympathetic activity, and the LF/HF ratio indicates sympathetic nerve activity [5].

Smartwatches may evaluate stress using HRV and contextual data. Advanced algorithms and machine learning methods analyze HRV data, compare it to established patterns, and predict stress levels based on the individual’s baseline and deviations [7,8]. Smartwatches and smartphone apps display HRV data and stress assessment findings. HRV trends, stress levels, and stress management suggestions are available [36]. Smartwatches simplify HRV monitoring; however, their accuracy and precision might vary. Sensor quality, skin contact, motion artifacts, and algorithm design affect HRV data dependability. Clinical-grade HRV analysis may need medical equipment and professional interpretation.

Smartwatches are beneficial for stress management, as they have key advantages. First, smartwatches monitor heart rate, HRV, and stress in real time. Awareness helps people recognize stress and reflect on its causes. Recognizing stress patterns helps people reduce stress [8]. Smartwatches provide continuous monitoring without extra equipment or difficult processes. They provide real-time heart rate, HRV, and stress data to users [37]. Continuous monitoring helps people track their progress, identify triggers, and make stress management changes. Smartwatches track stress patterns over time. Historical data and trends may help people understand their pressures, find patterns, and make educated lifestyle choices and stress reduction decisions [37,38]. These personalized insights may help people design personalized stress management plans.

Smartwatches guide users through stress-reduction activities like breathing and mindfulness. Step-by-step instructions, visual cues, and tactile feedback help consumers relax. This guidance emphasizes stress-reduction and stress management [14]. Smartwatches also measure sleep and exercise. A healthy lifestyle includes exercise, sleep, and stress management. Smartwatches encourage healthy behaviors that reduce stress and improve well-being [39,40].

Goal setting, progress monitoring, and reminders help smartwatches manage stress. These traits motivate and hold individuals accountable, encouraging them to actively reduce stress [41,42]. The smartwatch’s remarks and accomplishments may encourage stress management. Smartwatches may connect to many smartphone applications, health platforms, and services. This integration uses meditation apps, guided relaxation programs, and online support groups to manage stress holistically. Smartwatches are hubs for stress-reduction goods and services [30].

## 4. Comparison of Smartwatches and Compatible Mobile App for Stress Management

Apple is the leading smartwatch brand in the world with a market share of over 43% in 2023. Samsung is the second leading brand, with a market share of 12%. Fitbit and Garmin follow, with market shares of 8% and 7%, respectively. Polar has a market share of 3%. Other brands, such as Huawei, Noise, and imoo, make up the remaining market share [43].

Apple’s dominance in the smartwatch market is due to its strong brand reputation, its wide range of features, and its integration with the iPhone. Samsung’s smartwatches are popular for their stylish design and their compatibility with Samsung smartphones. Garmin smartwatches are known for their fitness tracking features, while Fitbit smartwatches are popular for their sleep tracking features. The global smartwatch market is expected to grow in the coming years, as more people adopt these devices for fitness tracking, health monitoring, and other purposes.

Here is a comparison of Apple, Garmin, Fitbit, Samsung, Polar smartwatch, EliteHRV, Welltory, and HRV4Training for HRV monitoring and biofeedback system for stress management (Table 1).

As you can see, all the smartwatches on this list offer HRV tracking and stress tracking. However, there are some differences in the features that they offer. For example, the Apple Watch has a Breathe app that can help someone relax and reduce their stress levels. Garmin watches have a stress score that can give an indication of someone’s overall stress levels. The Fitbit watches have a variety of stress management tools, such as guided breathing exercises. The Samsung watches have a Calm app that can provide someone with guided meditations and other relaxation techniques. And Polar watches have a serene breathing exercise that can help someone to slow down their breathing and increase HRV. Ultimately, the best smartwatch for HRV and stress management for consumers will depend on their individual needs and preferences. If someone is looking for a watch with a variety of stress management features, then Fitbit or Samsung watches may be a good option for them. If anyone is looking for a watch with a long battery life, then the Garmin watches may be a good option for them. And if anyone is looking for a watch with a high degree of accuracy, then the Apple Watch or Polar watches may be a good option for them.

In terms of accuracy, all the smartwatches on this list are considered to be good. However, it is important to note that the accuracy of HRV measurements can vary depending on several factors, such as the user’s body position, the ambient light, and the fit of the watch.

Finally, the battery life of the smartwatches on this list varies. The Apple Watch has a battery life of up to 18 h, the Garmin watches have a battery life of up to 14 days, the Fitbit watches have a battery life of up to 7 days, the Samsung watches have a battery life of up to 4 days, and the Polar watches have a battery life of up to 7 days.

In short, all the smartwatches and mobile apps listed above offer a variety of features for HRV monitoring and biofeedback. The best device for you will depend on the individual’s needs and budget. If you are looking for a device with a wide range of features, the Apple Watch, Garmin Forerunner 245, or Fitbit Sense are good options. If you are looking for a more affordable option, the Samsung Galaxy Watch 4 or Polar Vantage M2 are good choices. If you are looking for a device that is specifically designed for heart rate and HRV monitoring, EliteHRV, Welltory, or HRV4Training are good options.

## 5. Discussion

The present smartwatches in the market offer a range of health benefits, including activity tracking, heart rate monitoring, sleep tracking, stress management through HRV monitoring, blood oxygen level monitoring, ECG monitoring, fall detection and emergency alerts, GPS tracking, menstrual cycle tracking, and health data integration [52]. These features enable users to track their physical activity, monitor their heart health, analyze sleep patterns, manage stress levels, track respiratory health, detect abnormal heart rhythms, receive emergency assistance, track outdoor activities, monitor menstrual cycles, and consolidate health data for a holistic view of well-being [53]. However, it is important to note that the specific features and capabilities may vary across different smartwatch models and brands.

Smartwatches offer several health benefits through HRV (heart rate variability) monitoring for stress management. By utilizing HRV data, smartwatches can provide valuable insights into an individual’s stress levels and overall well-being. HRV, the variation in time intervals between heartbeats, is closely tied to the body’s autonomic nervous system and can serve as an objective measure of stress [8,9]. Smartwatches equipped with HRV sensors can continuously monitor and analyze these fluctuations, offering users a comprehensive understanding of their stress response patterns. It is reported that the HRV within-person reliability is more during sleep rather than wakefulness [54].

One of the primary health benefits of smartwatches through HRV for stress management is the ability to track and identify stress levels in real time [8]. By providing immediate feedback on HRV measurements, smartwatches can assist users in becoming more aware of their stress triggers and responses. This self-awareness can be a powerful tool for individuals seeking to manage their stress effectively. Additionally, smartwatches can offer stress management features such as guided breathing exercises or relaxation prompts based on HRV data [55]. These interventions can help users regulate their stress response and induce a state of relaxation and calmness.

Identifying discrepancies and gaps in the research for smartwatch comparisons for stress management reveals several important areas where the existing literature may be lacking. One prominent issue is the scarcity of extensive studies that directly compare the stress management capacities of different smartwatch models. Instead of conducting head-to-head comparisons, most studies tend to focus on specific features and the effectiveness of individual smartwatches. This limitation makes it challenging to determine which smartwatch models perform better in stress management scenarios [56].

Another significant factor contributing to the discrepancies is the variation in HRV algorithms used by different smartwatch brands and models. Each brand may employ its own algorithm to compute HRV and interpret stress levels. Consequently, there can be inconsistencies in the reported stress levels or scores across different smartwatches. This variance in algorithms complicates the direct evaluation of the efficiency and accuracy of various smartwatches in stress management [57].

The validity and reliability of HRV measurements and stress management functions also differ between smartwatch models. Some smartwatches may have undergone more rigorous scientific validation than others, resulting in variations in their accuracy and reliability in measuring and managing stress. This discrepancy further hinders the ability to compare and assess the performance of different smartwatches for stress management purposes [8,58].

Most of the existing research on smartwatches for stress management focuses on short-term effects and user engagement, neglecting the investigation of long-term effects and sustained user involvement. It is crucial to conduct longitudinal studies to determine the long-term impact of smartwatches on stress reduction and to assess whether users continue to engage with the stress management capabilities of their smartwatches over an extended period. This longitudinal research would provide valuable insights into the effectiveness and user acceptance of smartwatches in managing stress over time [59].

Furthermore, many studies examining smartwatches for stress management have primarily targeted specific demographics, such as athletes or individuals with certain health concerns. While these studies may provide valuable insights for these particular populations, they may not offer a comprehensive understanding of how smartwatches perform across diverse user groups. Therefore, it is essential to expand research efforts to include a broader range of user demographics to gain a more representative view of the effectiveness and applicability of smartwatches for stress management [60].

Lastly, the lack of standardization in stress measurement and assessment techniques across different smartwatches poses a significant challenge. The absence of consistent protocols and metrics makes it difficult to compare and generalize research findings. To address this issue, collaborative efforts among researchers, smartwatch manufacturers, and healthcare practitioners are necessary. Such collaborations can contribute to the development of standardized guidelines and metrics for evaluating the stress management capabilities of smartwatches, ensuring a more consistent and reliable basis for comparison [61].

In summary, the research on smartwatch comparisons for stress management exhibits several discrepancies and gaps. These include limited comparable studies, HRV algorithm variations, issues with validation and reliability, insufficient investigation of long-term effects and user engagement, a narrow focus on specific demographics, and the lack of standardization in stress measurement and assessment techniques. To fill these gaps, it is imperative to conduct more extensive comparative studies, particularly those directly evaluating stress management capacities using standardized protocols. Longitudinal trials involving diverse user populations and collaborations among stakeholders can help address these discrepancies and contribute to the development of standardized guidelines for evaluating the stress management capabilities of smartwatches.

## 6. Limitations and Future Directions

It is important to consider the limitations of HRV data obtained from smartwatches. One limitation is the potential for inaccuracies in the measurements. Smartwatches may not provide the same level of accuracy as medical-grade devices due to factors like device placement, motion artifacts, different algorithms, and sensor limitations. Also, proprietary algorithms are not subject to the same scrutiny as open-source algorithms. This means that there is a greater risk of bias and inaccuracy. Therefore, while smartwatches can offer valuable insights, they should not be considered as a substitute for professional medical advice or diagnostic tools [62,63].

Another limitation is the individual variability in HRV. Factors such as age, fitness level, and underlying health conditions can influence HRV, abnormally high or low HRV in different pathophysiological conditions and smartwatches may not always account for these individual differences [5,64]. Consequently, the stress assessments and recommendations provided by smartwatches may not always be tailored to an individual’s specific circumstances.

Moreover, contextual interpretation is crucial when analyzing HRV data. Elevated HRV readings do not necessarily indicate high stress, as other factors like physical activity or excitement can influence HRV as well [5,9]. It is essential to consider the broader context of an individual’s lifestyle and overall health when interpreting HRV data obtained from smartwatches.

Further, subjective perception plays a role in stress management. While HRV data can provide an objective measure of stress, it may not always align with an individual’s subjective experience of stress. Each person may have their own unique ways of perceiving and responding to stress, which HRV data alone may not fully capture [60].

Lastly, smartwatches with HRV monitoring capabilities offer valuable health benefits for stress management. They provide real-time feedback, promote self-awareness, and offer stress management interventions. However, it is important to recognize the limitations of HRV data obtained from smartwatches, including potential inaccuracies, individual variability, the need for contextual interpretation, and the subjective nature of stress perception. Integrating smartwatch data with professional medical advice and personalized care can ensure a comprehensive and effective approach to stress management [8,25].

The future direction of smartwatch applications for stress management is poised to bring significant advancements and hold valuable clinical implications. As technology continues to evolve, we can anticipate more sophisticated features and capabilities that enhance the effectiveness of stress management apps on smartwatches [65]. One area of focus is improving the accuracy of stress assessments by refining HRV measurements through advanced algorithms and sensor enhancements. By leveraging machine learning and artificial intelligence, smartwatches can analyze HRV data in real time, allowing for more precise and personalized stress monitoring [66].

The integration of machine learning algorithms can enable smartwatches to learn individual patterns, recognize unique stress response profiles, and provide tailored recommendations for stress management. These recommendations could include guided breathing exercises, mindfulness techniques, or adaptive relaxation prompts based on the user’s specific needs and preferences. Additionally, smartwatches could employ biofeedback mechanisms that provide real-time feedback on HRV and stress levels, helping users develop effective self-regulation techniques [66,67].

The clinical implications of these advancements are significant. Smartwatches equipped with robust stress management apps can empower individuals to proactively monitor and manage their stress levels. By gaining a deeper understanding of their stress responses and receiving personalized interventions, users may be able to mitigate the negative impact of chronic stress on their physical and mental health. This proactive approach to stress management could potentially reduce the risk of stress-related conditions such as cardiovascular disease, anxiety disorders, and burnout. Also, it should include more integration with their physician electronic records to have better individual interpretation and changes to the lifestyle and, in persistent chronic anxiety cases, directed cognitive and medical treatment [42,68,69].

Next, the integration of smartwatch data into clinical settings holds promise for healthcare professionals. With consent and appropriate privacy measures, clinicians could access patients’ stress data from their smartwatches to gain valuable insights into their stress levels and response patterns. These data can inform clinical assessments, treatment plans, and the evaluation of treatment efficacy. By incorporating smartwatch data into the clinical workflow, healthcare providers can offer more personalized and evidence-based stress management interventions [70].

In summary, the future of smartwatch applications for stress management looks promising. With advancements in technology, machine learning algorithms, and data analytics, smartwatches have the potential to become powerful tools for individuals to monitor and manage their stress levels effectively. Moreover, the clinical implications are significant, offering healthcare professionals valuable insights and tools to enhance patient care and improve stress management strategies [70,71].

## 7. Conclusions

Smartwatches equipped with HRV monitoring capabilities play a valuable role in various activities and stress management. These devices offer a range of features, including activity tracking, heart rate monitoring, sleep tracking, and stress management interventions based on HRV data. Its HRV data have provided very useful information of personal health regarding level of stress, self-management of stress factors and understanding of sleep and personal health. By providing real-time feedback, promoting self-awareness, and offering personalized stress management techniques, smartwatches empower individuals to monitor and manage their stress levels effectively. However, it is important to acknowledge the limitations of HRV data obtained from smartwatches, such as accuracy, individual variability, contextual interpretation, and the subjective nature of stress perception. Integrating smartwatch HRV data with professional medical advice and personalized care can ensure a comprehensive and effective approach to stress management. Overall, smartwatch HRV monitoring has the potential to enhance well-being by assisting individuals in achieving a healthier balance in their activities and stress levels.

## Figures and Tables

**Table 1 sensors-23-07314-t001:** Comparison among smartwatches such as Apple, Garmin, Fitbit, Samsung, Polar, and 3rd party app like Elite HRV, Welltory, HRV4Training with difference of biological measurement, HRV tracking watch and watch series, heart rate variability tracking, graph of HRV, stress tracking, stress tracking watch, stress management features, HRV biofeedback, personalization of real-time HRV biofeedback watch, time domain (RMSSD, SDNN) and frequency domain (HF, LF, LF/HF ratio), algorithms used, abnormal HR alerts, respiratory rate and depth, compatible with third-party apps, accuracy, battery life, user friendliness, price, and additional features.

Feature	Apple Watch	Garmin	Fitbit	Samsung	Polar	Elite HRV	Welltory	HRV4Training
Biological Measurement	HR, PA, Falls, Respiration, Sleep and ECG	HR, PA, Respiration and Sleep	HR, PA, Respiration and Sleep	HR, PA, Respiration and Sleep	HR, PA, Respiration and Sleep	HR, Respiration, Sleep, PA, Diet, Lifestyle factor	HR, Respiration, Stress score, Recovery score	HR, Respiration, Training load, Training effect
HRV Tracking watch and watch series	Apple watch 3, 4, 5, 6, 7, 8 and SE	Fenix 6, Epix (Gen 2) Forerunner 245/245 music/245S/245 S music, 945 LTE, 955/955 solar/955 plus, Instinct 2, Tactix 7, Venu 2	Sense, Versa 2, 3, Charge 4, 5, Inspire 2, Luxe	Galaxy 4 and 4 classic, Galaxy watch active 2, Galaxy watch 3, Fit, Fit 2	Polar Vantage V and V2, Grit X and X pro, M430, Ignite, Unite, polar 2	NA	NA	NA
Heart Rate variability Tracking	Yes	Yes	Yes	Yes	Yes	Yes	Yes	Yes
Graph of HRV	Yes	Yes	Yes	Yes	Yes	Yes	Yes	Yes
Stress Tracking	Yes	Yes	Yes	Yes	Yes	Yes	Yes	Yes
Stress Tracking watch	Apple watch 4, 5, 6, 7, 8	Fenix 6, Epix (Gen 2) Forerunner 245/245 music/245S/245 S music, 945 LTE, 955/955 solar/955 plus, Instinct 2, Tactix 7, Venu 2, Venu 2 plus	Sense, Versa 2, 3, Charge 4, 5, Inspire 2, Luxe	Galaxy 4 and 4 classic, Galaxy watch active 2, Galaxy watch 3, Fit, Fit 2	Polar Vantage V and V2, Grit X and X pro, M430, Ignite, Unite, Polar 2	NA	NA	NA
Stress Management features	Breathe app, Mindfulness app	Stress score, Body battery, Relaxation timer, Stress predictor	Stress management score, EDA scan, Relax app	Stress level, Breathe app, Stress management app	Nightly recharge, Serene app, Recovery pro features	Stress score, Breathing exercises, Meditation sessions, Sleep tracking, Recovery recommendations	Stress score, Breathing exercises, Meditation sessions, Sleep tracking, Recovery recommendations	Stress score, Breathing exercises, Recovery recommendations
HRV Biofeedback	Yes	Yes	Yes	Yes	Yes	Yes	Yes	Yes
Personalization of Real time HRV Biofeedback watch	Minimum	Medium (Garmin Forerunner 945, Garmin Venu 2, and Garmin Vivoactive 4)	Medium (Fitbit Sense and Fitbit Versa 3)	Minimum	Medium (Polar Vantage V2, Polar Grit X Pro, and Polar Ignite)	High	High	Medium
Time Domain (RMSSD, SDNN) and Frequency Domain (HF, LF, LF/HF ratio)	Yes	Yes	Yes	Yes	Yes	Yes	Yes	Yes
Algorithms used	Pan Tompkins algorithm, Hilbert transform	Pan Tompkins algorithm, Hilbert transform, Poincaré plot	Pan Tompkins algorithm, Hilbert transform	Pan Tompkins algorithm, Hilbert transform	Pan Tompkins algorithm, Hilbert transform	Pan Tompkins algorithm, Hilbert transform, Poincaré plot	Pan Tompkins algorithm, Hilbert transform	Pan Tompkins algorithm, Hilbert transform
Abnormal HR Alerts	Yes	Yes	Yes	Yes	Yes	Yes	Yes	Yes
Respiratory Rate and Depth	During Sleep	During Sleep	During Sleep and While awake	During Sleep and While awake	During Sleep and While awake	Needs Chest Strap	Can measure	Needs Chest Strap
Compatible with Third party Apps	Yes	Yes	Yes	Yes	Yes	No	Yes	Yes
Accuracy	Good	Good	Good	Good	Good	Excellent	Better than smartwatch	Better than smartwatch
Battery Life	Up to 18 h	From 24 h up to 14 days	Up to 7 days	From 40 h up to 4 days	From 30 h up to 7 days	Up to 24 h	Up to 12 h	Varies
User Friendliness	Good	Good	Good	Good	Good	Good	Good	Excellent
Price	Starting at $399	Starting at $299	Starting at $299	Starting at $249	Starting at $299	Starting from $0–30/mon	Starting from $13–79/mon	Starting at $9.99/mon
Additional Features	Activity tracking, sleep tracking, GPS, music, notifications, payments [44]	Activity tracking, multisport tracking, GPS, music, notifications, payments [45]	Sleep tracking, stress tracking, GPS, music, notifications, payments [46]	Activity tracking, health tracking, GPS, music, notifications, payments [47]	Activity tracking, GPS [48]	Heart rate variability analysis (HRV coherence, HRV spectral analysis) [49]	Stress management insights, heart rate variability analysis (HRV coherence, HRV spectral analysis) [50]	Heart rate variability analysis (HRV coherence, HRV spectral analysis) [51]

(HR = heart rate, PA = physical activity, HRV = heart rate variability, ECG = electrocardiogram, GPS = global positioning system, NA = not applicable).

## Data Availability

Not applicable.

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
