# Peer review of "The Future of Stress Management: Integration of Smartwatches and HRV Technology"

_sensors, 2023, doi:10.3390/s23177314_

Round 1
Reviewer 1 Report
The topic is timely and has the potential to inform scientists, clinicians and consumers. However, a lack of critical analysis of HRV methods, especially frequency domain based on FFT, weakens the paper. Furthermore, the description of methods for the determination of "accuracy" is inadequate.
The authors rely on the reviews of Kim and Pham when the primary sources may provide more depth.
While there is a large body of literature that employs FFT derived frequency domain data to draw inferences regarding sympathetic vs. parasympathetic balance, these claims have been criticized. Specifically, linear HRV analysis assumes that the underlying physiological mechanisms regulating heart rate variations are primarily linear and can be modeled using linear mathematical tools such as Fourier analysis (FFT). HRV signals are not stationary, and the relationships between frequency components are not linear and do not follow a Gaussian distribution. As noted by Hayana and Yuda, "There is no longer a convincing physiological basis justifying the assessment of cardiac sympathetic nervous functions whether by absolute or normalized LF power or LF/HF." Hayano, J., Yuda, E. Pitfalls of assessment of autonomic function by heart rate variability. J Physiol Anthropol 38, 3 (2019). https://doi.org/10.1186/s40101-019-0193-2
A serious shortcoming of many wearable/smartwatch apps is the use of proprietary algorithms which have not been independently validated and related to specific health outcomes and health benefits.
The authors state, "In terms of accuracy, all of the smartwatches on the list are considered "good." How was this determined? What measurements were made? Has this claim been independently verified?
The authors note that "the validity and reliability of HRV measurements and stress management functions also differ between smartwatch models." Are reliability and validity data available? Was the methodology used to assess validity and reliability for the various smartwatches similar enough to permit comparison between makes and models?
This paper is not suitable for publication as written. However, the topic is interesting, timely, and has potential. It may be considered after major revision.
Author Response
Dear Reviewer
Please see the attachment where we tried to clarify your query and did correction as well.

Reviewer 2 Report
please see attached file

Quality of English Language is acceptable, although there is scope for improvement. Emphasis on other logistical/organizational issues is given in previous comments.
Author Response
Dear Reviewer,
All issues arisen by you, those have been corrected. Please see attachment.
Thank you,
MS

Round 2
Reviewer 1 Report
Much improved. My issues were addressed.